# LVLM-Driven Attribute-Aware Modeling for Visible-Infrared Person Re-Identification

**Zhiqi Pang**[1]    **Lingling Zhao**[1]    **Junjie Wang**[2]    **Chunyu Wang**[1]*

[1] Harbin Institute of Technology, China
[2] Nanjing Medical University, China

zqpang98@gmail.com, zhaoll@hit.edu.cn, junjiehit@163.com, chunyu@hit.edu.cn

## Abstract

Visible-infrared person re-identification (VI-ReID) aims to match visible and infrared images of the same individual. Supervised VI-ReID (SVI-ReID) methods have achieved promising performance under the guidance of manually annotated identity labels. However, the substantial annotation cost severely limits their scalability in real-world applications. As a result, unsupervised VI-ReID (UVI-ReID) methods have attracted increasing attention. These methods typically rely on pseudo-labels generated by clustering and matching algorithms to replace manual annotations. Nevertheless, the quality of pseudo-labels is often difficult to guarantee, and low-quality pseudo-labels can significantly hinder model performance improvements. To address these challenges, we explore the use of attribute arrays extracted by a large vision-language model (LVLM) to enhance VI-ReID, and propose a novel LVLM-driven attribute-aware modeling (LVLM-AAM) approach. Specifically, we first design an attribute-aware reliable labeling strategy, which refines intra-modality clustering results based on image-level attributes and improves inter-modality matching by grouping clusters according to cluster-level attributes. Next, we develop an explicit-implicit attribute fusion module, which integrates explicit and implicit attributes to obtain more fine-grained identity-related text features. Finally, we introduce an attribute-aware contrastive learning module, which jointly leverages static and dynamic text features to promote modality-invariant feature learning. Extensive experiments conducted on VI-ReID datasets validate the effectiveness of the proposed LVLM-AAM and its individual components. LVLM-AAM not only significantly outperforms existing unsupervised methods but also surpasses several supervised methods.

## 1 Introduction

Person re-identification (ReID) [45, 2, 4, 57, 34, 25, 10, 35] focuses on identifying images of a specific person from a large-scale gallery. To advance intelligent surveillance systems across various time periods, visible-infrared ReID (VI-ReID) [31, 46, 42, 22, 9] was introduced, aiming to match visible and infrared images of the same person. While supervised VI-ReID (SVI-ReID) methods [43, 51, 7, 50] have shown promising performance on multiple datasets, they heavily rely on manually annotated identity labels in the training set. However, manually annotating data for VI-ReID tasks is an extremely labor-intensive process. As a result, increasing attention has been given to unsupervised VI-ReID (UVI-ReID) [21, 38, 32, 36, 37, 29, 30, 33]. These UVI-ReID methods typically begin by clustering image features extracted by an image encoder to generate intra-modality pseudo-labels. They then perform inter-modality matching to generate inter-modality pseudo-labels. These intra-modality and inter-modality pseudo-labels serve as supervisory signals to replace manual labels, thus reducing the limitations imposed by the high cost of annotation.

---

*Corresponding author: Chunyu Wang

39th Conference on Neural Information Processing Systems (NeurIPS 2025).

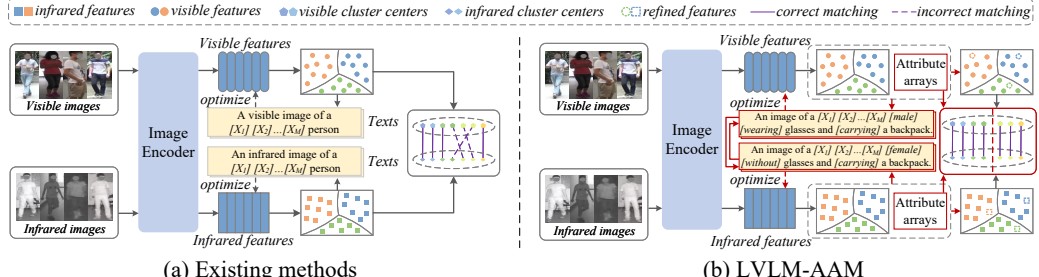

(a) Existing methods      (b) LVLM-AAM

Figure 1: The differences between existing methods and LVLM-AAM are highlighted with red lines. Different feature colors indicate different pseudo-labels. (a) The texts are entirely derived from pseudo-labels without enhancing the quality of the pseudo-labels. Moreover, these texts are only utilized for optimization within their corresponding modality. (b) LVLM-AAM leverages attribute arrays provided by an LVLM to simultaneously enhance intra-modality pseudo-labels, inter-modality matching (pseudo-labels), and texts. Furthermore, the text semantics can be mutually transferred between modalities during the optimization process. We define a correct match as two clusters from different modalities that contain images of the same identity, and an incorrect match otherwise.

However, the quality of pseudo-labels generated by clustering algorithms is largely constrained by the performance of the pretrained image encoder, and the global-level inter-modality matching often leads to mutual interference, resulting in cascading errors [32]. Although CLIP-based UVI-ReID methods [3] attempt to leverage the pretrained CLIP model to obtain text semantics as additional supervision signals, as shown in Figure 1a, two critical limitations still remain: (1) The texts are constructed based on pseudo-labels generated by clustering algorithms, thus inherently carrying similar supervision signals without fundamentally improving the pseudo-labels. (2) This method typically utilizes text semantics only to optimize features within the corresponding modality, focusing on enhancing intra-modality identity discrimination, but without explicitly assisting the image encoder in learning modality-invariant features.

Inspired by the powerful fine-grained vision-language understanding capability of the large vision-language model (LVLM) [1], we attempt to leverage attribute arrays extracted by an LVLM to improve VI-ReID, as illustrated in Figure 1b. We propose a novel LVLM-driven attribute-aware modeling (LVLM-AAM) method to address the aforementioned two problems. To tackle the first issue, we first employ an LVLM to extract attribute arrays (Gender, Upper, Lower, Glasses, and Backpack) for each image in the training set, which are referred to as *explicit attributes* in the following descriptions. Then, we design an attribute-aware reliable labeling (ARL) strategy, which consists of attribute-aware refinement (AR) and attribute-aware matching (AM). Specifically, AR refines intra-modality clustering results based on image-level attribute arrays, while AM groups clusters based on cluster-level attribute arrays to enhance inter-modality matching. Subsequently, we develop an explicit-implicit attribute fusion (EAF) module, which fuses implicit attributes (text embeddings) and explicit attributes to obtain more fine-grained identity-related text features. To address the second issue, we further propose attribute-aware contrastive learning (AAC), which not only computes dynamic text features based on static text features, but also optimizes with both static and dynamic features to enhance modality-invariant feature learning.

It is worth noting that, since LVLM-AAM leverages supervision information (i.e., attribute arrays) extracted from an LVLM, it may not be considered a fully UVI-ReID method. In other words, the primary goal of this work is to explore the effectiveness of utilizing an LVLM to advance the practical application of UVI-ReID – specifically, to improve recognition performance while maintaining low manual annotation costs. The contributions of this work are summarized as follows:

- We explore the use of attribute arrays extracted by an LVLM to improve VI-ReID and propose a novel LVLM-AAM method, which leverages attribute arrays to refine pseudo-labels and text semantics for enhanced modality-invariant feature learning.

- We design an ARL strategy and an EAF module. The former refines intra-modality and inter-modality pseudo-labels based on image-level and cluster-level attributes, respectively, while the latter utilizes attribute arrays to generate fine-grained text features.

- We develop an AAC module, which computes dynamic text features based on static text features from both modalities, and optimizes with both static and dynamic features to further enhance modality-invariant learning.
- Extensive experiments conducted on VI-ReID datasets validate the effectiveness of the proposed LVLM-AAM and its components. LVLM-AAM not only significantly outperforms existing unsupervised methods but also surpasses several supervised methods.

## 2   Related Work

**Visible-Infrared Person Re-Identification.**   Given an infrared image of a person, VI-ReID [31, 46, 42, 22, 9] aims to retrieve the corresponding visible image from a large-scale gallery, and vice versa. Early studies primarily focused on the supervised VI-ReID (SVI-ReID) [43, 51, 7, 53] setting, where manual annotations were used to guide the learning process and reduce the impact of modality gap. Although SVI-ReID methods have demonstrated promising recognition capabilities, they are constrained by the high cost of manual annotations. As a result, increasing attention has been directed toward the unsupervised VI-ReID (UVI-ReID) setting. For instance, H2H [21], as one of the early UVI-ReID approaches, first pretrains the image encoder on a manually labeled visible dataset [54], and then performs unsupervised learning on a visible-infrared dataset. ADCA [38] further eliminates the reliance on manual annotations for pretraining by first conducting homogeneous learning within each modality, followed by heterogeneous learning through inter-modality matching. Building upon ADCA, PGM [32] introduces a graph matching strategy to globally establish inter-modality positive clusters. Among recent methods, SDCL [37] enhances model optimization by exploring the relationships between shallow and deep features within the Transformer architecture [5, 14], providing abundant supervisory signals. PCLHD [30] introduces hard prototypes to supply diverse supervisory signals for optimization. Although existing UVI-ReID approaches have made significant efforts to obtain diverse and reliable supervision, these methods primarily rely on image feature distances or similarities. In contrast to previous work, we go beyond purely image-based features by leveraging attribute arrays extracted by the LVLM as additional supervisory signals to improve VI-ReID performance.

**Vision-Language Models.**   Powered by large-scale pretraining, vision-language models (VLMs) [27, 56] have demonstrated strong vision-language understanding capabilities and achieved competitive performance across various downstream tasks in computer vision. CLIP [27], as a representative work in the VLMs domain, typically consists of an image encoder and a text encoder. Given an input image-text pair, CLIP [27] aims to predict their similarity. Subsequent research, such as CoOp [56], enhances the flexibility of CLIP [27] by learning a set of task-specific text embeddings for each image category in downstream tasks. Inspired by the success of VLMs, researchers in the ReID community have also begun exploring VLMs-based approaches. For instance, CLIP-ReID [18] leverages text semantics by learning a set of text embeddings for each identity to assist the image encoder in extracting identity-related features. TVI-LFM [15] utilizes text descriptions of visible images to augment the corresponding infrared images of the same identity, thereby improving visible-infrared retrieval performance. In contrast to the aforementioned supervision methods, CCLNet [3] in the UVI-ReID field learns a set of text embeddings for each intra-modality pseudo-label obtained by clustering, and uses the optimized text embeddings as supervisory signals to enhance intra-cluster compactness. However, existing UVI-ReID approaches face two key limitations: (1) the optimized text embeddings provide supervision signals similar to the original pseudo-labels, without substantially improving the quality of the pseudo-labels; (2) the optimized text embeddings are solely used to promote intra-modality identity learning, without explicitly assisting the model in learning modality-invariant features. To address these issues, our proposed LVLM-AAM method leverages attribute arrays to refine intra-modality and inter-modality pseudo-labels and jointly utilizes both explicit and implicit attributes to promote modality-invariant feature learning.

## 3   The Proposed Method

### 3.1   Task Formulation and Method Overview

In the UVI-ReID task, we are provided with an unlabeled training set consisting of a visible image set $\{x_i^v\}_{i=1}^{N^v}$ and an infrared image set $\{x_i^r\}_{i=1}^{N^r}$. Our goal is to train an image encoder capable of

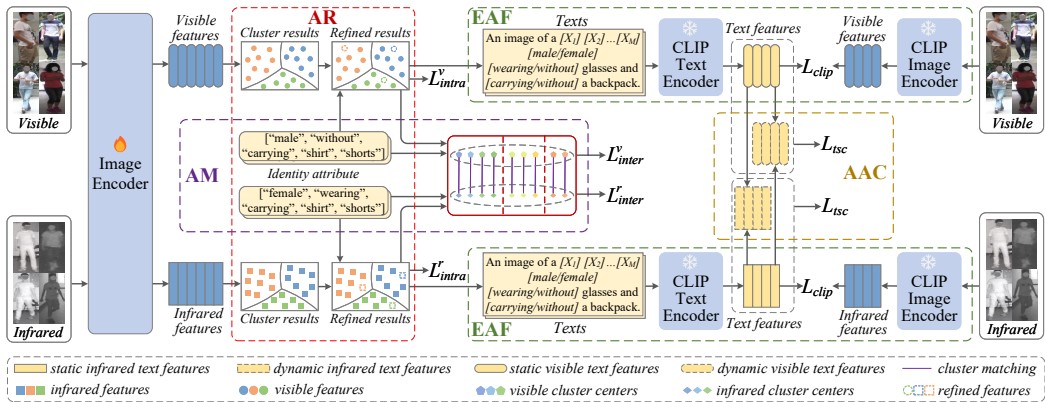

Figure 2: Illustration of our LVLM-AAM. The different colors of the features represent different pseudo-labels, while different shapes denote different modality labels.

extracting modality-invariant and identity-discriminative features. Before optimizing the image encoder, we utilize an LVLM [1] to extract a five-value attribute array for each image in $\{x_i^v\}_{i=1}^{N^v}$ and $\{x_i^r\}_{i=1}^{N^r}$. Specifically, we first prompt the LVLM [1] to respond in the following format: "Gender: [male/female]," "Glasses: [wearing/without]," "Backpack: [carrying/without]," "Upper: [clothing type]," and "Lower: [clothing type]." The detailed prompt can be found in Supplementary Material Section S.I. Subsequently, we arrange the extracted attribute values in the order of Gender, Glasses, Backpack, Upper, and Lower, obtaining a five-value attribute array for each image (e.g., ["male", "without", "carrying", "shirt", "shorts"]). In the following descriptions, we refer to these arrays as *explicit attributes* to distinguish them from *implicit attributes* (i.e., text embeddings).

As shown in Figure 2, the proposed LVLM-driven attribute-aware modeling (LVLM-AAM) method first employs an attribute-aware reliable labeling (ARL) strategy, which consists of an attribute-aware refinement (AR) module and an attribute-aware matching (AM) module, to obtain reliable intra-modality and inter-modality pseudo-labels, respectively. Meanwhile, an explicit-implicit attribute fusion (EAF) module is introduced, which leverages both explicit and implicit attributes along with a pretrained CLIP [27] model consisting of a text encoder and an image encoder to generate fine-grained text features. Finally, LVLM-AAM adopts an attribute-aware contrastive learning (AAC) module to generate dynamic text features, thereby guiding the image encoder to learn identity-discriminative and modality-invariant features.

### 3.2 Attribute-Aware Reliable Labeling

**Attribute-Aware Refinement.** We first input the visible image set $\{x_i^v\}_{i=1}^{N^v}$ and the infrared image set $\{x_i^r\}_{i=1}^{N^r}$ into the image encoder to obtain the visible feature set $\{f_i^v\}_{i=1}^{N^v}$ and the infrared feature set $\{f_i^r\}_{i=1}^{N^r}$. Then, we apply the DBSCAN algorithm [6] to perform clustering within both the visible and infrared modalities to obtain intra-modality pseudo-labels. Next, for any given cluster (pseudo-label), we determine the mode of attribute array values at each position across all attribute arrays within the cluster. In each cluster, these five mode values are then aggregated to formulate a cluster-level attribute array. From the perspective of ensemble learning [17, 41], the mode-based cluster-level attribute array better reflects the overall characteristics of the cluster than a single image-level attribute array. Therefore, we use the cluster-level attribute array to refine the images within the cluster by excluding those that deviate significantly from the cluster-level attribute array. Specifically, we identify image-level attribute arrays that differ from the cluster-level attribute array by more than $\eta$ values as outliers and exclude the corresponding images. Ultimately, we obtain the refined clustering results (pseudo-labels), which are used for intra-modality contrastive learning.

**Attribute-Aware Matching.** Existing inter-modality matching methods [32, 3, 30] typically perform matching at a global level. However, global cluster matching often leads to cascading errors due to mutual interference among clusters. Fortunately, the introduction of cluster-level attributes provides a foundation for more fine-grained inter-modality matching. Specifically, we group clusters based on the first three attribute values in their cluster-level attribute arrays. For example, clusters

with the first three attributes as "male," "without," and "carrying" are grouped together, while those with "male," "wearing," and "carrying" are placed in a different group. In this work, we obtain a total of eight groups, with each group containing both visible and infrared clusters. During the inter-modality matching process, we perform progressive graph matching [32] within each group to generate inter-modality pseudo-labels. The detailed matching process is provided in Supplementary Material Section S.II.

We use all attributes in the attribute-aware refinement module for outlier detection to align with a realistic perceptual principle: evaluating an object from more dimensions (attributes) is generally more accurate and comprehensive than doing so from fewer dimensions. We restrict attribute-aware matching to the first three attributes because they have countable value spaces, which ensures that each group contains both visible and infrared clusters.

### 3.3 Intra-Inter Modality Contrastive Learning

**Intra-Modality Contrastive Learning.** Within each modality, we compute the cluster centers based on the refined clustering results. For example, the center of the $p$-th cluster in the visible modality is defined as:

$$c_p^v = \sum_{i=1}^{N_p^v} f_i^v, \tag{1}$$

where $f_i^v$ denotes an image feature within the cluster, and $N_p^v$ represents the number of images in the cluster. The center of the $p$-th cluster in the infrared modality, denoted as $c_p^r$, is computed in a similar manner. Finally, we introduce the intra-modality contrastive loss to encourage the image encoder to learn identity-discriminative features. For example, for any image feature $f_i^v$ in the visible modality, the intra-modality contrastive loss is defined as:

$$L_{intra}^v = -\log \frac{\exp(f_i^v \cdot c_p^{v\mathrm{T}}/\tau)}{\sum_{q=1}^{C^v} \exp(f_i^v \cdot c_q^{v\mathrm{T}}/\tau)}, \tag{2}$$

where $c_p^v$ denotes the center of the cluster to which $f_i^v$ belongs, $C^v$ is the total number of clusters in the visible modality at the current epoch, and $\tau$ is the temperature hyperparameter. The intra-modality contrastive loss for the infrared modality, denoted as $L_{intra}^r$, is computed in a similar manner. Thus, the total intra-modality contrastive loss is defined as:

$$L_{intra} = L_{intra}^v + L_{intra}^r. \tag{3}$$

Attribute-aware refinement and intra-modality contrastive learning are iteratively performed, and the final intra-modality pseudo-labels are preserved.

**Inter-Modality Contrastive Learning.** We optimize the image encoder based on the inter-modality pseudo-labels (matches) obtained from attribute-aware matching. For example, for any visible feature $f_i^v$, the inter-modality contrastive loss is defined as:

$$L_{inter}^v = -\log \frac{\exp(f_i^v \cdot c_p^{r\mathrm{T}}/\tau)}{\sum_{q=1}^{C^r} \exp(f_i^v \cdot c_q^{r\mathrm{T}}/\tau)}, \tag{4}$$

where $c_p^r$ denotes the center of the infrared cluster matched to the cluster to which $f_i^v$ belongs, $C^r$ is the number of clusters in the infrared modality at the current epoch, and $\tau$ is a temperature hyperparameter. Similarly, the inter-modality contrastive loss $L_{inter}^r$ for infrared features can be defined in the same way. Following the alternate cross contrastive learning scheme [32], the overall inter-modality contrastive loss is defined as:

$$L_{inter} = \begin{cases} L_{inter}^v, epoch\%2 = 0 \\ L_{inter}^r, epoch\%2 = 1 \end{cases}, \tag{5}$$

where $epoch$ represents the index of the current epoch.

## 3.4 Explicit-Implicit Attribute Fusion

After the iterative execution of attribute-aware refinement and intra-modality contrastive learning, we assign to each cluster a text containing learnable text embeddings "$[X_1]\,[X_2]\ldots\ldots[X_M]$" in the format "An image of a $[X_1]\,[X_2]\ldots\ldots[X_M]\,[male/female]\,[wearing/without]$ glasses and $[carrying/without]$ a backpack." Here, $M$ represents the number of learnable text embeddings, and $[male/female]$, $[wearing/without]$, and $[carrying/without]$ correspond to the first three attribute values from the cluster-level attribute array of the respective cluster. By incorporating both explicit and implicit attributes, we enrich the semantic content of the text.

Subsequently, we input the image and its corresponding text into the pretrained CLIP [27] image and text encoders, respectively, to obtain the image features $f_p^s$ and text features $t_p^s$. Following existing optimization strategies, we freeze the pretrained CLIP [27] image and text encoders and introduce the CLIP contrastive loss [3] to optimize the learnable text embeddings:

$$L_{clip} = L_{i2t} + L_{t2i}, \tag{6}$$

$$L_{i2t} = -\log \frac{\exp(f_p^s \cdot t_p^{s\,\mathrm{T}})}{\sum_{q=1}^{B} \exp(f_p^s \cdot t_q^{s\,\mathrm{T}})}, \tag{7}$$

$$L_{t2i} = -\frac{1}{|P_p^s|} \sum_{f_p^s \in P_p^s} \log \frac{\exp(t_p^s \cdot f_p^{s\,\mathrm{T}})}{\sum_{q=1}^{B} \exp(t_p^s \cdot f_q^{s\,\mathrm{T}})}, \tag{8}$$

where $s \in \{v, r\}$ denotes the index for the visible or infrared modality, and $f_p^s$ and $t_p^s$ represent the image and text features with the same pseudo-label. $B$ refers to the batch size, and $P_p^t$ is the set of image features in the batch that share the same pseudo-label as $t_p^s$. Finally, we refer to the converged text features as the *static text features*.

## 3.5 Attribute-Aware Contrastive Learning

Although the static text features contain identity-related semantic information, they are modality-dependent. Therefore, to leverage the text features for promoting modality-invariant learning, we first obtain dynamic text features based on the static text features following attribute-aware matching. For example, for the $p$-th cluster in the visible modality, its dynamic text feature is defined as:

$$\hat{t}_p^v = (1 - \alpha)t_p^v + \alpha t_p^r, \tag{9}$$

where $t_p^v$ and $t_p^r$ represent the static text features of the cluster and its matching cluster in the infrared modality, respectively. $\alpha$ is the weight hyperparameter. The dynamic text features in the infrared modality are computed using a similar approach. The dynamic text feature incorporates information from both the visible and infrared modalities, and thus tends to be more modality-invariant compared to the static text features. Moreover, since the dynamic text feature is derived from two static text features that share the same inter-modality pseudo-label, it also retains identity-related information.

Subsequently, we introduce a text semantic contrastive loss to promote modality-invariant learning. For any image feature $f_p^s$, the text semantic contrastive loss is defined as:

$$L_{tsc} = -\log \frac{\exp(f_p^s \cdot \hat{t}_p^{s\,\mathrm{T}})}{\sum_{t_q^s \in Q^s \cup \hat{t}_p^s} \exp(f_p^s \cdot t_q^{s\,\mathrm{T}})}, \tag{10}$$

where $\hat{t}_p^s$ represents the dynamic text feature corresponding to $f_p^s$, and $Q^s$ denotes the set of all static text features in the modality to which $f_p^s$ belongs.

In summary, the total loss used to optimize the image encoder is defined as:

$$L_{total} = L_{intra} + \lambda_{inter} L_{inter} + \lambda_{tsc} L_{tsc}, \tag{11}$$

where $\lambda_{inter}$ and $\lambda_{tsc}$ are the weight hyperparameters for $L_{inter}$ and $L_{tsc}$, respectively. The overall algorithmic procedure is provided in Supplementary Material Section S.III.

# 4 Experiment

## 4.1 Datasets and Evaluation Metrics

We evaluate our method on the SYSU-MM01 [31], RegDB [23] and LLCM [52] datasets. SYSU-MM01 consists of images from 491 identities captured by four visible cameras and two near-infrared cameras. Following existing methods [3, 37], a total of 22,258 visible images and 11,909 infrared images from 395 identities are used for training. The query set and gallery consist of infrared and visible images, respectively, from the remaining 96 identities. RegDB contains 412 identities, with each identity having 10 visible images and 10 thermal infrared images. Following existing protocols [3, 37], we use images from 206 identities for training and the remaining 206 identities for testing. LLCM is collected under complex low-light conditions, making it a more challenging dataset compared to the previous two. It contains 46,767 bounding boxes of 1,064 identities captured by 9 cameras.

Following existing methods [32, 37], we use cumulative matching characteristics (CMC), mean average precision (mAP), and mean inverse negative penalty (mINP) to evaluate performance.

## 4.2 Implementation Details

The image encoder of LVLM-AAM is based on a pretrained ResNet-50 [13] and consists of two branches to separately handle inputs from the visible and infrared modalities. For the learnable text embeddings, we set $M = 4$. All images are resized to 288×144, and random flipping, random grayscale conversion [19], channel augmentation [47], and random erasing [55] are applied as data augmentation. We set the batch size $B$ to 128. In each iteration, we randomly select 8 clusters from each modality, and sample 16 images from each cluster. We use DBSCAN [6] to perform intra-modality clustering, where the distance threshold and the minimum number of samples are set to 0.6 and 4, respectively, on SYSU-MM01 [31], and to 0.3 and 4 on RegDB [23]. We adopt the Adam optimizer [16] for model training. Homogeneous learning (i.e., Eq. 3) is performed for 50 epochs, followed by an update of the learnable text embeddings (i.e., Eq. 6) over another 50 epochs. Finally, heterogeneous learning (i.e., Eq. 11) is conducted for an additional 50 epochs. The initial learning rate is set to 0.00035, and it decays 10 times every 20 epochs. The temperature hyperparameter $\tau$ is set to 0.05. For attribute-aware refinement (AR), we set $\eta = 2$. For attribute-aware contrastive learning (AAC), we set $\alpha = 0.5$. Regarding the weight hyperparameters for $L_{inter}$ and $L_{tsc}$, we set $\lambda_{inter} = 0.5$ and $\lambda_{tsc} = 0.5$. An analysis of the sensitivity of LVLM-AAM to the hyperparameters $\eta$ and $\lambda_{tsc}$ can be found in the Supplementary Material Section S.IV. The experiments are conducted on four NVIDIA GeForce RTX 4090 GPUs. The LVLM inference is only required once during the training phase to extract identity attributes and is not needed during the testing phase. Therefore, the computational cost associated with LVLM inference does not affect the inference speed of the trained ReID model during testing.

## 4.3 Comparison with the State-of-the-art Methods

As shown in Table 1, we compare the proposed LVLM-AAM with existing methods on SYSU-MM01 (both All Search and Indoor Search) and RegDB (Visible to Thermal). Among existing UVI-ReID methods, SDCL [37] and DLM [48] have achieved strong performance on SYSU-MM01 and RegDB, respectively. Our proposed LVLM-AAM surpasses both SDCL [37] and DLM [48] in overall performance across both datasets. Specifically, on SYSU-MM01 (All Search), LVLM-AAM outperforms SDCL [37] by 2.09%, 0.26%, and 1.23% in terms of Rank-1 accuracy, mAP, and mINP, respectively. On RegDB (Visible to Thermal), LVLM-AAM achieves improvements of 2.70%, 1.13%, and 2.05% over DLM [48] on the same three metrics. This is mainly because LVLM-AAM not only effectively leverages attribute arrays provided by the LVLM to obtain reliable pseudo-labels, but also jointly utilizes explicit and implicit attributes to further promote modality-invariant feature learning. SVI-ReID methods rely on manually annotated identity labels, which are inaccessible to LVLM-AAM and require significantly higher human effort compared to the attribute arrays used by LVLM-AAM. Encouragingly, LVLM-AAM achieves superior overall performance on both datasets compared to early SVI-ReID methods (e.g., DDAG [44], AGW [45], and MCLNet [12]), and demonstrates competitive results against more recent methods (e.g., FMCNet [51] and DART [39]). Moreover, on the RegDB dataset, the Rank-1 accuracy of LVLM-AAM is already comparable to that of the latest SVI-ReID methods (e.g., SAAI [7] and STAR-ReID [26]). These results further validate the

Table 1: Comparison with state-of-the-art methods on the SYSU-MM01 and RegDB datasets. The best performances among UVI-ReID methods are highlighted in bold, while performances of SVI-ReID methods that are lower than those of LVLM-AAM are indicated in italics.

| | Methods | SYSU-MM01 | | | | | | RegDB | | |
| | | All Search | | | Indoor Search | | | Visible to Thermal | | |
| | | Rank-1 | mAP | mINP | Rank-1 | mAP | mINP | Rank-1 | mAP | mINP |
|---|---|---|---|---|---|---|---|---|---|---|
| UVI-ReID | H2H [21] | 30.15 | 29.40 | - | - | - | - | 23.81 | 18.87 | - |
| | ADCA [38] | 45.51 | 42.73 | 28.29 | 50.60 | 59.11 | 55.17 | 67.20 | 64.05 | 52.67 |
| | CHCR [24] | 47.72 | 45.34 | - | - | - | - | 69.31 | 64.74 | - |
| | CCLNet [3] | 54.03 | 50.19 | - | 56.68 | 65.12 | - | 69.94 | 65.53 | - |
| | PGM [32] | 57.27 | 51.78 | 34.96 | 56.23 | 62.74 | 58.13 | 69.48 | 65.41 | - |
| | GUR [36] | 63.51 | 61.63 | 47.93 | 71.11 | 76.23 | 72.57 | 73.91 | 70.23 | 58.88 |
| | SDCL [37] | **64.49** | **63.24** | **51.06** | **71.37** | **76.90** | **73.50** | 86.91 | 78.92 | 62.83 |
| | PCLHD [30] | 64.4 | 58.7 | - | 69.5 | 74.4 | - | 84.3 | 80.7 | - |
| | PCAL [40] | 57.94 | 52.85 | 36.90 | 60.07 | 66.73 | 62.09 | 86.43 | 82.51 | 72.33 |
| | DLM [48] | 62.15 | 58.42 | 43.70 | 67.31 | 72.86 | 68.89 | **87.55** | **82.83** | **71.93** |
| SVI-ReID | DDAG [44] | *54.75* | *53.02* | *39.62* | *61.02* | *67.98* | *62.61* | *69.34* | *63.46* | *49.24* |
| | AGW [45] | *47.50* | *47.65* | *35.30* | *54.17* | *62.97* | *59.23* | *70.05* | *66.37* | *50.19* |
| | MCLNet [12] | *65.40* | *61.98* | *47.39* | *72.56* | *76.58* | *72.10* | *80.31* | *73.07* | *57.39* |
| | FMCNet [51] | *66.34* | *62.51* | - | *68.15* | *74.09* | - | *89.12* | 84.43 | - |
| | DART [39] | 68.72 | 66.29 | 53.26 | *72.52* | *78.17* | *74.94* | *83.60* | *75.67* | *60.60* |
| | SGIEL [8] | 77.12 | 72.33 | - | 82.07 | 82.95 | - | 92.18 | 86.59 | - |
| | MUN [49] | 76.24 | 73.81 | - | 79.42 | 82.06 | - | 95.19 | 87.15 | - |
| | SAAI [7] | 75.90 | 77.03 | - | 83.20 | 88.01 | - | 91.07 | 91.45 | - |
| | IDKL [28] | 81.42 | 79.85 | - | 87.14 | 89.37 | - | 94.72 | 90.19 | - |
| | STAR-ReID [26] | 82.93 | 80.47 | - | 88.04 | 89.58 | - | 91.89 | 93.31 | - |
| | **LVLM-AAM** | **66.58** | **63.50** | **52.29** | **72.97** | **78.65** | **75.21** | **90.25** | **83.96** | **73.98** |

Table 2: Comparison with state-of-the-art methods on the LLCM dataset.

| Methods | Reference | Visible to Infrared | | Infrared to Visible | |
| | | Rank-1 | mAP | Rank-1 | mAP |
|---|---|---|---|---|---|
| CCLNet [3] | MM'23 | 45.3 | 49.9 | 39.3 | 45.3 |
| PGM [32] | CVPR'23 | 44.4 | 48.6 | 38.4 | 44.2 |
| SDCL [37] | CVPR'24 | 46.9 | 52.4 | 43.4 | 48.2 |
| SCA-RCP [20] | TKDE'24 | 29.1 | 33.3 | 22.3 | 28.0 |
| LVLM-AAM | Ours | **52.2** | **57.3** | **46.0** | **51.7** |

effectiveness of LVLM-AAM and highlight the potential of replacing costly manual annotations with automatically extracted attribute arrays.

We further compare LVLM-AAM with state-of-the-art unsupervised methods on the LLCM dataset. As shown in Table 2, our method outperforms existing methods in both testing scenarios of LLCM. For instance, compared to SDCL [37], LVLM-AAM achieves a more significant performance gain on LLCM than on SYSU-MM01. Moreover, despite SCA-RCP [20] utilizing camera labels, LVLM-AAM still demonstrates a substantial advantage. This is because LLCM presents greater complexity compared to SYSU-MM01 and RegDB, making it more challenging for existing methods to obtain reliable pseudo-labels. In contrast, LVLM-AAM effectively leverages attribute arrays from the LVLM to enhance the reliability of pseudo-labels and utilizes text semantics to facilitate model optimization, thereby achieving superior performance over existing methods. These experimental results not only validate the superiority of the proposed LVLM-AAM but also demonstrate its strong generalization capability.

## 4.4 Ablation Study

In this section, we evaluate the effectiveness of attribute-aware refinement (AR), attribute-aware matching (AM), explicit-implicit attribute fusion (EAF), and attribute-aware contrastive learning (AAC). As shown in Table 3, the baseline adopts the same image encoder and CLIP-based architecture as LVLM-AAM. The key difference lies in that the baseline does not incorporate AR, AM, or EAF, and replaces AAC with the image-to-text contrastive loss (ITC) [3], which utilizes text features containing only implicit attributes to assist the optimization of the image encoder. Four ablation

Table 3: Ablation study on the SYSU-MM01 and RegDB datasets.

| Methods | AR | AM | EAF | ITC | AAC | SYSU-MM01(All Search) | | | RegDB(Visible to Thermal) | | |
|---|---|---|---|---|---|---|---|---|---|---|---|
| | | | | | | Rank-1 | mAP | mINP | Rank-1 | mAP | mINP |
| Baseline | | | | ✓ | | 58.52 | 52.89 | 35.27 | 72.55 | 68.59 | 56.90 |
| A1 | ✓ | | | ✓ | | 60.85 | 56.03 | 41.26 | 75.63 | 71.28 | 60.05 |
| A2 | ✓ | ✓ | | ✓ | | 61.36 | 57.40 | 43.18 | 78.34 | 73.57 | 62.17 |
| A3 | ✓ | ✓ | ✓ | ✓ | | 62.51 | 59.26 | 46.11 | 80.29 | 75.16 | 65.23 |
| A4 | ✓ | ✓ | | | ✓ | 64.59 | 61.89 | 50.69 | 86.57 | 79.85 | 70.29 |
| A5 | ✓ | ✓ | ✓ | | ✓ | 66.58 | 63.50 | 52.29 | 90.25 | 83.96 | 73.98 |

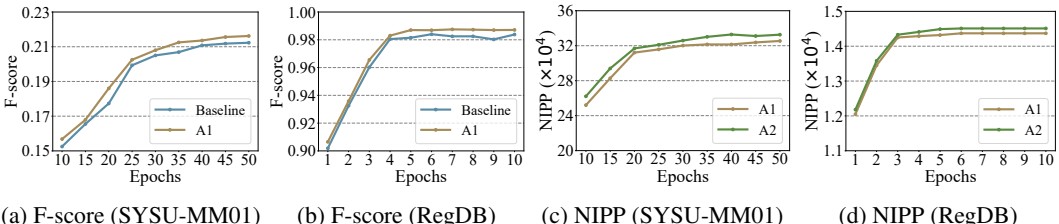

(a) F-score (SYSU-MM01)    (b) F-score (RegDB)    (c) NIPP (SYSU-MM01)    (d) NIPP (RegDB)

Figure 3: Statistical results of F-score and NIPP.

variants (A1, A2, A3, and A4), along with the complete LVLM-AAM (A5), progressively introduce AR, AM, EAF, and AAC based on the baseline.

**Effectiveness of AR and AM.** As shown in Table 3, introducing AR in A1 leads to noticeable performance improvements over the baseline on both SYSU-MM01 and RegDB. For example, on SYSU-MM01 (All Search), Rank-1, mAP, and mINP are improved by 2.33%, 3.14%, and 5.99%, respectively, while on RegDB (Visible to Thermal), the three metrics are increased by 3.08%, 2.69%, and 3.15%, respectively. To further evaluate the effectiveness of AR, we assess the F-score [11] of the intra-modality pseudo-labels. A higher F-score [11] indicates greater accuracy of the pseudo-labels. As shown in Figure 3a and Figure 3b, A1 achieves a significantly higher F-score compared to the baseline, verifying that AR can effectively leverage attribute arrays to enhance the reliability of pseudo-labels and thereby improve model performance. Building upon A1, A2 introduces AM and achieves further performance improvements. For example, on SYSU-MM01 (All Search), Rank-1, mAP, and mINP increase by 0.51%, 1.37%, and 1.92%, respectively. On RegDB (Visible to Thermal), the three metrics improve by 2.71%, 2.29%, and 2.12%, respectively. In addition, we analyze the number of inter-modality positive pairs (NIPP) obtained by A1 and A2. Specifically, two images are considered an inter-modality positive pair if they share the same inter-modality pseudo-label, the same ground-truth identity label, and different modality labels. Generally, a higher NIPP indicates more accurate inter-modality matching. As shown in Figure 3c and Figure 3d, A2 increases NIPP compared to A1. This confirms that AM can enhance model performance by improving the accuracy of inter-modality matching.

**Effectiveness of EAF and AAC.** EAF introduces explicit attributes to enrich the text semantics. As shown in Table 3, A3 incorporates EAF based on A2 and achieves further performance improvements on both SYSU-MM01 and RegDB. In the VI-ReID task, due to the modality gap, the feature distance between inter-modality positive pairs is typically much larger than that between intra-modality positive pairs. As shown in Figure 4, compared to A2, A3 slightly reduces the feature distance between inter-modality positive pairs. This improvement can be attributed to EAF introducing explicit, modality-invariant, and identity-relevant attributes to enrich the text semantics, thereby effectively enhancing modality-invariant feature learning. A4 builds upon A2 by introducing AAC, which involves computing dynamic text features and replacing the image-to-text contrastive loss (ITC) [3] with the text semantic contrastive loss (Eq. 10). As shown in Table 3, A4 achieves significant performance improvements over A2 after introducing AAC. For example, on SYSU-MM01 (All Search), Rank-1, mAP, and mINP improve by 3.23%, 4.49%, and 7.51%, respectively. As illustrated in Figure 4, A4 significantly reduces the feature distance between inter-modality positive pairs compared to A2. This is because AAC incorporates inter-modality matching information into the

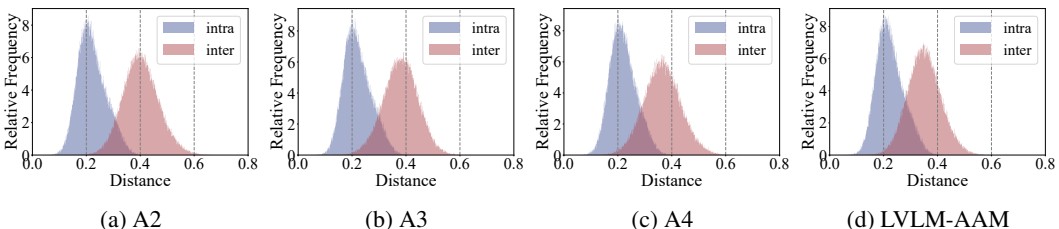

Figure 4: The feature distance distributions of intra-modality and inter-modality positive pairs for A2, A3, A4, and LVLM-AAM.

dynamic text features and encourages image features to approach the dynamic text features during optimization, thereby enhancing modality-invariant feature learning.

Furthermore, we observe that LVLM-AAM (A5) outperforms all of the aforementioned ablation variants and further reduces the feature distance between inter-modality positive pairs compared to A3 and A4. This validates that AR, AM, EAF, and AAC can be organically integrated to enhance the model's performance in cross-modality scenarios.

## 5 Conclusion and Limitations

In this paper, we propose an LVLM-driven attribute-aware modeling (LVLM-AAM) method to improve VI-ReID. Ablation studies validate the effectiveness of each module in LVLM-AAM. Specifically, attribute-aware reliable labeling, which comprises attribute-aware refinement and attribute-aware matching, effectively leverages attribute arrays to enhance the reliability of both intra-modality and inter-modality pseudo-labels. Explicit-implicit attribute fusion utilizes attribute arrays to acquire fine-grained identity-related text features, while attribute-aware contrastive learning promotes modality-invariant learning by integrating static and dynamic text features. Comparative experimental results demonstrate the superiority of LVLM-AAM, which not only significantly outperforms existing unsupervised methods and earlier supervised approaches but also competes with state-of-the-art supervised methods in certain scenarios.

Essentially, this paper represents an early exploration of applying the LVLM to UVI-ReID, with the core contribution being the preliminary validation of the effectiveness of attribute arrays extracted by the LVLM in UVI-ReID. However, a thorough analysis of a broader range of attribute arrays has not been conducted, which could serve as a potential direction for future research. Additionally, the eight groupings manually set in attribute-aware matching are a very preliminary exploration, and exploring more diverse or flexible strategies in the future could yield even more promising results.

## Acknowledgements

This work is supported by National Key Research and Development Program of China (Grant no. 2023YFC3305003, 2023YFC3305000)

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
