# OpenReview forum: "LVLM-Driven Attribute-Aware Modeling for Visible-Infrared Person Re-Identification"
_NeurIPS.cc/2025/Conference — NeurIPS 2025 poster_

### Official Review · Reviewer_nF4X · 2025-06-13

**Clarity:** 3
**Significance:** 3
**Originality:** 4
**Rating:** 5
**Confidence:** 5

**Summary:**

This paper proposes a novel LVLM-driven Attribute-Aware Modeling (LVLM-AAM) method for unsupervised visible-infrared person re-identification (UVI-ReID). In contrast to existing approaches, the authors leverage attribute arrays extracted from a Large Vision-Language Models (LVLM) not only to refine pseudo-labels but also to incorporate attribute text semantics as supervisory signals for model optimization. Furthermore, both static and dynamic text features are integrated to enhance feature learning. Experimental results on the SYSU-MM01 and RegDB datasets demonstrate that the proposed LVLM-AAM achieves competitive performance and significantly outperforms a range of state-of-the-art methods.

**Questions:**

1. To what extent might the design of the attribute arrays be biased toward the SYSU-MM01 and RegDB datasets? More specifically, does the methodology involving attribute arrays impose constraints on the generalizability of the approach? The authors’ response to this concern will significantly influence my decision regarding the final evaluation score.
2. Have the authors measured the computational cost associated with extracting attribute arrays using the LVLM? For instance, it would be helpful to report metrics such as memory consumption and inference time. Is this process computationally expensive?
3. The performance of the proposed model appears to be closely tied to that of the underlying LVLM. Have the authors experimented with different LVLMs? Does achieving strong performance require considerable effort in selecting a high-performing LVLM?

**Ethical Concerns:**

["NO or VERY MINOR ethics concerns only"]

**Final Justification:**

This paper proposes to utilize attribute arrays extracted from an LVLM to improve Unsupervised Visible-Infrared Person Re-Identification, which is a novel and meaningful attempt to introduce visual-language into the person re-identification scenario. Please incorporate experimental results and discussion into the revised version.

**Limitations:**

Yes

**Quality:**

3

**Strengths And Weaknesses:**

Strengths:
1. This paper introduces a novel perspective by leveraging attribute arrays extracted from an LVLM to enhance UVI-ReID. This approach not only reduces the reliance on manually annotated data but also yields significant performance gains. Given that many existing methods in this field primarily focus on extracting knowledge from smaller pretrained models, the proposed strategy offers meaningful insights for future research.
2. The methodology is well-motivated and clearly articulated. The attribute-aware reliable labeling strategy effectively mitigates the common issue of low-quality pseudo-labels in unsupervised learning, while the attribute-aware contrastive learning module successfully exploits semantic information to facilitate the modality-invariant feature learning.
3. The proposed method achieves competitive results, outperforming not only existing unsupervised approaches but also several supervised ones.

Weaknesses:
1. The most critical concern lies in the potential overfitting of the proposed LVLM-AAM framework to the SYSU-MM01 and RegDB datasets, due to the handcrafted nature of the attribute arrays. If these arrays were tailored with prior knowledge of these benchmarks, the generalizability of the method to other datasets may be limited.
2. Although incorporating an LVLM has the potential to enhance recognition performance, it inevitably introduces additional computational overhead, which may hinder deployment in resource-constrained scenarios.
3. The performance of the model appears to be heavily dependent on the quality of the underlying LVLM. Selecting and configuring a high-performing LVLM can be both time-consuming and computationally expensive, which could pose practical challenges for broader adoption.

---

> ### Author Rebuttal · Authors · 2025-07-29
>
> We thank the reviewer for the constructive and detailed feedback.
>
> ---
>
> > *Weakness1 and Question1: To what extent might the design of the attribute arrays be biased toward the SYSU-MM01 and RegDB datasets? More specifically, does the methodology involving attribute arrays impose constraints on the generalizability of the approach? The authors’ response to this concern will significantly influence my decision regarding the final evaluation score.*
>
> **Response:**
> Thank you for your comment. The attribute arrays are not specifically designed for the SYSU-MM01 and RegDB datasets. Instead, they consist of general and commonly observed human attributes. Therefore, the attribute arrays do not limit the generalizability of the proposed method.
>
> To evaluate the generalization ability of the proposed LVLM-AAM, we assess its performance on the LLCM dataset. The comparison between the proposed LVLM-AAM and existing state-of-the-art unsupervised methods on the LLCM dataset is presented in the Table below, where we report the Rank-1 accuracy for each method. Our method outperforms existing methods in both testing scenarios of LLCM. For instance, compared to SDCL [3], LVLM-AAM achieves a more significant performance gain on LLCM than on SYSU-MM01. Moreover, despite SCA-RCP [4] utilizing camera labels, LVLM-AAM still demonstrates a substantial advantage. This is because LLCM presents greater complexity compared to SYSU-MM01 and RegDB, making it more challenging for existing methods to obtain reliable pseudo-labels. In contrast, LVLM-AAM effectively leverages attribute arrays from the LVLM to enhance the reliability of pseudo-labels and utilizes text semantics to facilitate model optimization, thereby achieving superior performance over existing methods. These experimental results not only validate the superiority of the proposed LVLM-AAM but also demonstrate its strong generalization capability.
>
> |    Methods    | LLCM (Visible to Infrared) | LLCM (Infrared to Visible) |
> |-------------|:--------------------------:|:---------------------------:|
> |  CCLNet [1]   |           45.3             |            39.3             |
> |   PGM [2]     |           44.4             |            38.4             |
> |  SDCL [3]     |           46.9             |            43.4             |
> | SCA-RCP [4]   |           29.1             |            22.3             |
> |  LVLM-AAM     |           52.2             |            46.0             |
>
> ---
>
> > *Weakness2 and Question2: Have the authors measured the computational cost associated with extracting attribute arrays using the LVLM? For instance, it would be helpful to report metrics such as memory consumption and inference time. Is this process computationally expensive?*
>
> **Response:**
> In our experiments, we use Qwen2.5-VL-32B-Instruct, a moderately sized model, to extract attribute arrays. In terms of memory consumption, Qwen2.5-VL-32B-Instruct requires approximately 66 GB of GPU memory, which can be easily accommodated by our hardware setup (i.e., four NVIDIA GeForce RTX 4090 GPUs) used for training the ReID model. The inference time of Qwen2.5-VL-32B-Instruct is also generally acceptable. For example, extracting attributes for all training images in the SYSU-MM01 dataset typically takes around 15 hours.
>
> In summary, the computational cost associated with the LVLM is not prohibitively high. Moreover, since the LVLM is only used during the training phase to extract attributes, it does not affect the inference speed of the trained ReID model during testing.
>
> ---
>
> > *Weakness3 and Question3: The performance of the proposed model appears to be closely tied to that of the underlying LVLM. Have the authors experimented with different LVLMs? Does achieving strong performance require considerable effort in selecting a high-performing LVLM?*
>
> **Response:**
> Thank you for your question. The performance of the proposed method is indeed influenced by the capability of the underlying LVLM. Specifically, we experiment with several LVLM models, including Qwen2.5-VL-3B-Instruct, Qwen2.5-VL-7B-Instruct, Qwen2.5-VL-32B-Instruct, and Qwen2.5-VL-72B-Instruct.
>
> In each trial, we first use the LVLM to extract identity-related attributes, which are then used to guide the training of the ReID model. The entire process associated with Qwen2.5-VL-3B-Instruct and Qwen2.5-VL-7B-Instruct takes less than one day, while the process with Qwen2.5-VL-32B-Instruct and Qwen2.5-VL-72B-Instruct takes less than two days. As a result, selecting a high-performing LVLM can typically be completed within a week, which is often shorter than the overall hyperparameter tuning time of the ReID model. Therefore, identifying a suitable LVLM does not require a substantial amount of effort.
>
> Finally, we select Qwen2.5-VL-32B-Instruct, which offers a good trade-off between computational cost and performance, to extract attributes, ensuring the reliability of the attributes used in our method.
>
> We will include the above experiments and analysis in the revised manuscript and supplementary materials.
>
> ---
>
> [1] Zhong Chen, Zhizhong Zhang, Xin Tan, Yanyun Qu, and Yuan Xie. Unveiling the power of clip in unsupervised visible-infrared person re-identification. In ACM MM, pages 3667-3675, 2023.
>
> [2] Zesen Wu and Mang Ye. Unsupervised visible-infrared person re-identification via progressive graph matching and alternate learning. In CVPR, pages 9548-9558, 2023.
>
> [3] Bin Yang, Jun Chen, and Mang Ye. Shallow-deep collaborative learning for unsupervised visible-infrared person re-identification. In CVPR, pages 16870-16879, 2024.
>
> [4] Zhiyong Li, Haojie Liu, Xiantao Peng, and Wei Jiang. Inter-intra modality knowledge learning and clustering noise alleviation for unsupervised visible-infrared person re-identification. IEEE TKDE, 36(8): 3934-3947, 2024.

---

> > ### Comment · Reviewer_nF4X · 2025-08-01
> >
> > Thank the authors for the detailed response, particularly the additional validation regarding generalizability, which has largely addressed my concerns. In light of these improvements, I have decided to raise my rating to Accept.

---

> > > ### Author Response · Authors · 2025-08-04
> > >
> > > Thank you very much for your thoughtful feedback and for your updated evaluation. We truly appreciate your recognition, and your support means a great deal to us and our work.

---

### Official Review · Reviewer_B8pp · 2025-06-27

**Clarity:** 3
**Significance:** 4
**Originality:** 4
**Rating:** 6
**Confidence:** 5

**Summary:**

This paper proposes an approach to enhance unsupervised visible-infrared person re-identification by leveraging large vision-language models (LVLMs). Specifically, the authors utilize attribute arrays extracted from an LVLM to refine both intra-modality clustering and inter-modality matching. Furthermore, attribute information is embedded into the text to enrich semantic guidance. The method also incorporates both dynamic and static text features to facilitate modality-invariant feature learning. Experimental results on two datasets demonstrate the superiority of the proposed method, outperforming existing unsupervised methods and even some supervised methods. The effectiveness of each component is further supported by comprehensive ablation studies.

**Questions:**

1. Does the proposed method demonstrate generalizability? Specifically, have the authors evaluated the proposed method on broader datasets such as LLCM [1] to assess its robustness and effectiveness?
2. What is the underlying rationale for the joint use of dynamic and static text features in promoting modality-invariant feature learning? A more detailed explanation or theoretical justification would enhance the clarity of this design choice.
3. During the inter-modality contrastive learning stage, the proposed method appears to rely on previously saved pseudo-labels rather than generating new ones through clustering algorithms. This seems to diverge from several existing methods [2, 3] that dynamically update pseudo-labels in each epoch. Could the authors clarify the motivation behind this decision?

If the authors are able to provide well-reasoned responses to the above concerns, I would be willing to raise my rating.

[1] Yukang Zhang, Hanzi Wang. Diverse embedding expansion network and low-light cross-modality benchmark for visible-infrared person re-identification. In CVPR, 2153-2162, 2023.
[2] Bin Yang, Mang Ye, Jun Chen, Zesen Wu. Augmented Dual-Contrastive Aggregation Learning for Unsupervised Visible-Infrared Person Re-Identification. In ACM MM, 2843–2851, 2022.
[3] Zesen Wu, Mang Ye. Unsupervised Visible-Infrared Person Re-Identification via Progressive Graph Matching and Alternate Learning. In CVPR, 9548-9558, 2023.

**Ethical Concerns:**

["NO or VERY MINOR ethics concerns only"]

**Final Justification:**

The validation regarding generalizability, as well as the explanations on modality-invariant feature learning and pseudo-labels, are effective and satisfactory. I consider this to be an interesting paper with solid contributions, and I have accordingly raised my rating to Strong Accept.

**Limitations:**

Yes

**Quality:**

3

**Strengths And Weaknesses:**

Strengths:
1. This work appears to be one of the early attempts to explore the use of Large Vision-Language Models (LVLMs) in the context of unsupervised visible-infrared person re-identification. The proposed method effectively leverages attribute arrays generated by an LVLM to enhance performance, making the study both meaningful and novel.
2. The experimental results are convincing. In particular, Figures 3 and 4 in the ablation study provide strong evidence supporting the effectiveness of the key components introduced by the authors.
3. The manuscript is generally well written, with a clear presentation and a logical organization of sections. The inclusion of the Algorithmic Procedure helps elucidate the proposed method.
Weaknesses:
1. The generalizability of the proposed method is not sufficiently demonstrated, as experiments are limited to only two datasets, with RegDB being relatively small in scale.
2. The authors claim that static text features are modality-dependent. However, they do not provide a clear explanation as to why combining dynamic and static text features facilitates the modality-invariant feature learning.

---

> ### Author Rebuttal · Authors · 2025-07-29
>
> We thank the reviewer for the constructive and detailed feedback.
>
> ---
>
> > *Weakness1 and Question1: Does the proposed method demonstrate generalizability? Specifically, have the authors evaluated the proposed method on broader datasets such as LLCM to assess its robustness and effectiveness?*
>
> **Response:**
> Thank you for your valuable comments. To evaluate the generalization ability of the proposed LVLM-AAM, we assess its performance on the LLCM dataset. The comparison between the proposed LVLM-AAM and existing state-of-the-art unsupervised methods on the LLCM dataset is presented in the Table below, where we report the Rank-1 accuracy for each method. Our method outperforms existing methods in both testing scenarios of LLCM. For instance, compared to SDCL [3], LVLM-AAM achieves a more significant performance gain on LLCM than on SYSU-MM01. Moreover, despite SCA-RCP [4] utilizing camera labels, LVLM-AAM still demonstrates a substantial advantage. This is because LLCM presents greater complexity compared to SYSU-MM01 and RegDB, making it more challenging for existing methods to obtain reliable pseudo-labels. In contrast, LVLM-AAM effectively leverages attribute arrays from the LVLM to enhance the reliability of pseudo-labels and utilizes text semantics to facilitate model optimization, thereby achieving superior performance over existing methods. These experimental results not only validate the superiority of the proposed LVLM-AAM but also demonstrate its strong generalization capability.
>
> |    Methods    | LLCM (Visible to Infrared) | LLCM (Infrared to Visible) |
> |-------------|:--------------------------:|:---------------------------:|
> |  CCLNet [1]   |           45.3             |            39.3             |
> |   PGM [2]     |           44.4             |            38.4             |
> |  SDCL [3]     |           46.9             |            43.4             |
> | SCA-RCP [4]   |           29.1             |            22.3             |
> |  LVLM-AAM     |           52.2             |            46.0             |
>
> ---
>
> > *Weakness2 and Question2: What is the underlying rationale for the joint use of dynamic and static text features in promoting modality-invariant feature learning? A more detailed explanation or theoretical justification would enhance the clarity of this design choice.*
>
> **Response:**
> Since the static text features are learned separately within each modality, they typically contain modality-specific information. As shown in Equation (9) (with $\alpha = 0.5$), the dynamic text feature incorporates equal contributions from both the visible and infrared modalities, and thus tends to be more modality-invariant compared to the static text features. Therefore, in Equation (10), by encouraging image features to align with the dynamic text feature while being pushed away from the static text features of the same modality, our method has the potential to promote modality-invariant feature learning.
>
> Moreover, since the dynamic text feature is derived from two static text features that share the same inter-modality pseudo-label across the visible and infrared modalities, it not only possesses modality-invariant information but also retains identity-related information. Therefore, incorporating the dynamic text feature has the potential to facilitate both modality-invariant feature learning and identity-discriminative feature learning.
>
> ---
>
> > *Question3: During the inter-modality contrastive learning stage, the proposed method appears to rely on previously saved pseudo-labels rather than generating new ones through clustering algorithms. This seems to diverge from several existing methods that dynamically update pseudo-labels in each epoch. Could the authors clarify the motivation behind this decision?*
>
> **Response:**
> Thank you for your critical question. Our inter-modality contrastive learning indeed relies on previously saved pseudo-labels rather than generating new ones through clustering algorithms. This design choice is based on two main considerations. First, we observe in our experiments that the pseudo-labels generated by clustering during the inter-modality contrastive learning stage are largely consistent with those obtained in the intra-modality contrastive learning stage, with only a small portion differing. Therefore, using the saved pseudo-labels allows us to maintain pseudo-label accuracy while significantly reducing the computational cost during training. Second, in the explicit-implicit attribute fusion stage, we optimize the learnable text embeddings based on the pseudo-labels saved from the intra-modality contrastive learning stage and use the resulting text features to guide the optimization of the image encoder. To ensure consistency between the supervision signals provided by the text features and the pseudo-labels, we choose not to regenerate pseudo-labels during the inter-modality contrastive learning stage. This strategy helps maintain training stability while also improving efficiency.
>
> We will include the above experiments and analysis in the revised manuscript and supplementary materials.
>
> ---
>
> [1] Zhong Chen, Zhizhong Zhang, Xin Tan, Yanyun Qu, and Yuan Xie. Unveiling the power of clip in unsupervised visible-infrared person re-identification. In ACM MM, pages 3667-3675, 2023.
>
> [2] Zesen Wu and Mang Ye. Unsupervised visible-infrared person re-identification via progressive graph matching and alternate learning. In CVPR, pages 9548-9558, 2023.
>
> [3] Bin Yang, Jun Chen, and Mang Ye. Shallow-deep collaborative learning for unsupervised visible-infrared person re-identification. In CVPR, pages 16870-16879, 2024.
>
> [4] Zhiyong Li, Haojie Liu, Xiantao Peng, and Wei Jiang. Inter-intra modality knowledge learning and clustering noise alleviation for unsupervised visible-infrared person re-identification. IEEE TKDE, 36(8): 3934-3947, 2024.

---

> > ### Comment · Reviewer_B8pp · 2025-08-02
> >
> > The experimental results regarding generalizability are convincing. The authors' explanations of modality-invariant feature learning and the use of pseudo-labels are also reasonable. These responses have effectively addressed my concerns. I consider this to be an interesting paper with solid contributions.

---

> > > ### Author Response · Authors · 2025-08-05
> > >
> > > Thank you for your positive feedback and recognition. Your recognition of our contributions is very important to us and greatly appreciated.

---

### Official Review · Reviewer_pR1y · 2025-06-30

**Clarity:** 3
**Significance:** 2
**Originality:** 2
**Rating:** 5
**Confidence:** 5

**Summary:**

This paper presents LVLM-AAM, an attribute-aware modeling method for unsupervised visible-infrared person re-identification. By leveraging a large vision-language model (LVLM) to extract attributes like gender and clothing, the method refines pseudo-labels and enhances modality-invariant learning. It features attribute-aware reliable labeling, attribute fusion, and contrastive learning, outperforming existing unsupervised and some supervised methods on SYSU-MM01 and RegDB datasets.

**Questions:**

1. The paper should analyze the effectiveness of EAF and AAC in a fully supervised setting.
2. It lacks an analysis of hyper-parameters such as alpha, lambda, and eta.
3. The title should include "Unsupervised" unless the paper aims to demonstrate the method's effectiveness in both unsupervised and supervised Visible-Infrared Person Re-Identification scenarios, in which case experiments in the supervised context should be added.

**Ethical Concerns:**

["NO or VERY MINOR ethics concerns only"]

**Final Justification:**

I appreciate the detailed response, which has resolved my questions. I am satisfied and will accept it.

**Limitations:**

yes

**Quality:**

2

**Strengths And Weaknesses:**

Strengths :
The experimental results on SYSU-MM01 and RegDB datasets achieve state-of-the-art performance.

Weaknesses:
The attributes generated by LVLM are identical to those created manually, differing only in efficiency. Consequently, the AR and AM modules are straightforward and not novel, with no direct connection to LVLM. The use of attributes itself is not innovative. Therefore, the main contribution remains in the cross-modality alignment components, specifically EAF and AAC, which require further analysis to validate their effectiveness.

---

> ### Author Rebuttal · Authors · 2025-07-29
>
> We thank the reviewer for the constructive and detailed feedback.
>
> ---
>
> > *Weaknesses: The attributes generated by LVLM are identical to those created manually, differing only in efficiency. Consequently, the AR and AM modules are straightforward and not novel, with no direct connection to LVLM. The use of attributes itself is not innovative. Therefore, the main contribution remains in the cross-modality alignment components, specifically EAF and AAC, which require further analysis to validate their effectiveness.*
>
> **Response:**
> Thank you for your comment. As you point out, using LVLM-generated attributes to replace manually annotated ones indeed improves efficiency, which is a key factor in promoting the practical deployment of ReID systems.
>
> The core contribution and novelty of this paper lie in leveraging supervision information (i.e., attribute arrays) generated by an LVLM to improve the performance of unsupervised visible-infrared person re-identification. This represents a novel research direction that, to the best of our knowledge, has not been explored in existing visible-infrared person re-identification methods. The other three reviewers also describe this as a “highly novel direction.” Building upon this direction, we design four key modules: attribute-aware refinement (AR), attribute-aware matching (AM), explicit-implicit attribute fusion (EAF), and attribute-aware contrastive learning (AAC). Among them, AR and AM utilize LVLM-derived attribute arrays to effectively enhance the reliability of pseudo-labels and improve model performance. These components are absent from prior methods and demonstrate how the powerful capabilities of LVLMs can be effectively connected to and leveraged within the ReID task.
>
> Following your suggestion, we evaluate the effectiveness of EAF and AAC under the fully supervised setting. Relevant details are provided in the response below.
>
> ---
>
> > *Question1: The paper should analyze the effectiveness of EAF and AAC in a fully supervised setting.*
>
> **Response:**
> Thank you very much for your guidance. We have evaluated the effectiveness of EAF and AAC under the fully supervised setting. In this setting, the baseline method uses ground-truth identity labels instead of the pseudo-labels generated by the proposed attribute-aware reliable labeling strategy. The detailed setup follows CLIP-VIReID+MsPL [1]. We then incorporate EAF and AAC into the baseline method individually to assess their contributions.
>
> The ablation analysis of EAF and AAC under fully supervised settings is shown in the Table below, where we report the Rank-1 accuracy for each variant. EAF effectively improves model performance by embedding attribute arrays from the LVLM into text prompts. Specifically, Baseline+EAF improves Rank-1 accuracy over the Baseline by 3.38% and 5.27% on SYSU-MM01 (All Search and Indoor Search, respectively), and by 7.04% on RegDB (Visible to Thermal). Furthermore, AAC introduces dynamic text features to further enhance performance. Compared to Baseline+EAF, Baseline+EAF+AAC achieves additional Rank-1 improvements of 1.05% and 1.12% on SYSU-MM01 (All Search and Indoor Search, respectively), and 0.93% on RegDB (Visible to Thermal). Similarly, Baseline+AAC also outperforms the Baseline. These results validate the effectiveness of both EAF and AAC under the fully supervised setting.
>
> |      Methods       | SYSU-MM01 (All Search) | SYSU-MM01 (Indoor Search) | RegDB (Visible to Thermal) |
> |------------------|:----------------------:|:--------------------------:|:---------------------------:|
> |     Baseline       |         72.89          |           78.67           |            87.60           |
> |   Baseline+EAF     |         76.27          |           83.94           |            94.64           |
> |   Baseline+AAC     |         75.63          |           81.79           |            91.53           |
> | Baseline+EAF+AAC   |         77.32          |           85.06           |            95.57           |
>
> ---
>
> > *Question2: It lacks an analysis of hyper-parameters such as alpha, lambda, and eta.*
>
> **Response:**
> Thank you for your comment. We set $\alpha = 0.50$ to ensure fairness between modalities. Specifically, in the dynamic text feature, the information from the infrared and visible modalities should contribute equally to maintain a certain degree of modality invariance. In addition, we also experiment with $\alpha = 0.25$ and $\alpha = 0.75$, and find that the model achieves better performance when $\alpha = 0.50$ compared to the other two settings. This result supports our design choice.
>
> We set $\lambda_{inter} = 0.50$ by following the configuration used in existing methods [2]. In addition, we experiment with a range of values including [0.00, 0.25, 0.50, 0.75, 1.00], and observe that the model achieves the best performance when $\lambda_{inter} = 0.50$. This may be attributed to the fact that inter-modality pseudo-labels are generally less reliable than intra-modality pseudo-labels. Therefore, while $L_{inter}$ contributes to performance improvement, its weight $\lambda_{inter}$ should be significantly lower than the weight of $L_{intra}$, which is set to 1.
>
> For $\lambda_{tsc}$ and $\eta$, we provide experimental results and analysis in Section S.IV of the supplementary material.
>
> ---
>
> > *Question3: The title should include "Unsupervised" unless the paper aims to demonstrate the method's effectiveness in both unsupervised and supervised Visible-Infrared Person Re-Identification scenarios, in which case experiments in the supervised context should be added.*
>
> **Response:**
> Thank you for your insightful comment. We did not include the term “Unsupervised” in the title because LVLM-AAM leverages supervision information (i.e., attribute arrays) extracted from an LVLM, and thus may not be strictly categorized as a fully unsupervised method. As stated in Lines 62-66 of the manuscript: “since LVLM-AAM leverages supervision information (i.e., attribute arrays) extracted from an LVLM, it may not be considered a fully UVI-ReID method. In other words, the primary goal of this work is to explore the effectiveness of utilizing an LVLM to advance the practical application of UVI-ReID -- specifically, to improve recognition performance while maintaining low manual annotation costs.”
>
> Following your suggestion, we evaluate the proposed method in the supervised visible-infrared person re-identification scenario and compare its performance with existing supervised methods. Specifically, under the fully supervised setting, we replace the pseudo-labels generated by the proposed attribute-aware reliable labeling with the ground-truth identity labels. The comparison between the proposed LVLM-AAM and existing state-of-the-art supervised methods on the SYSU-MM01 and RegDB datasets is presented in the Table below, where we report the Rank-1 accuracy for each method. Although the proposed LVLM-AAM is not specifically designed for the supervised scenario, it still achieves competitive performance under the supervised setting. For example, LVLM-AAM performs comparably to state-of-the-art supervised methods such as SAAI [3] and CSDN [1] on SYSU-MM01, and even outperforms existing supervised methods on RegDB. This is attributed to the ability of LVLM-AAM to effectively leverage the attribute array from the LVLM to assist model optimization. These results validate the effectiveness of the proposed LVLM-AAM in the supervised scenario.
>
> |     Methods     | SYSU-MM01 (All Search) | SYSU-MM01 (Indoor Search) | RegDB (Visible to Thermal) |
> |---------------|:----------------------:|:--------------------------:|:---------------------------:|
> |   SAAI [3]      |         75.90          |           83.20           |            91.07           |
> |   IDKL [4]      |         81.42          |           87.14           |            94.72           |
> | STAR-ReID [5]   |         82.93          |           88.04           |            91.89           |
> |   CSDN [1]      |         76.70          |           84.50           |            95.40           |
> |   LVLM-AAM      |         77.32          |           85.06           |            95.57           |
>
> We will include the above experiments and analysis in the revised manuscript and supplementary materials.
>
> ---
>
> [1] Xiaoyan Yu, Neng Dong, Liehuang Zhu, Hao Peng, and Dapeng Tao. Clip-driven semantic discovery network for visible-infrared person re-identification. IEEE Transactions on Multimedia, 2025.
>
> [2] Zesen Wu and Mang Ye. Unsupervised visible-infrared person re-identification via progressive graph matching and alternate learning. In CVPR, pages 9548-9558, 2023.
>
> [3] Xingye Fang, Yang Yang, and Ying Fu. Visible-infrared person re-identification via semantic alignment and affinity inference. In ICCV, pages 11270-11279, 2023.
>
> [4] Kaijie Ren and Lei Zhang. Implicit discriminative knowledge learning for visible-infrared person re-identification. In CVPR, pages 393-402, 2024.
>
> [5] Yuxuan Qiu, Liyang Wang, Wei Song, Jiawei Liu, Zhiping Shi, and Na Jiang. Advancing visible-infrared person re-identification: Synergizing visual-textual reasoning and cross-modal feature alignment. IEEE TIFS, 2025.

---

### Official Review · Reviewer_nERK · 2025-07-01

**Clarity:** 3
**Significance:** 3
**Originality:** 3
**Rating:** 4
**Confidence:** 5

**Summary:**

This paper proposes LVLM-AAM, a novel Large Vision-Language Model (LVLM)-driven Attribute-Aware Modeling approach for unsupervised visible-infrared person re-identification (UVI-ReID). The core idea is to leverage attribute arrays extracted by an LVLM to enhance the quality of cluster-based pseudo-labels and guide modality-invariant feature learning. Extensive experiments demonstrate significant performance improvements over existing unsupervised methods and even some supervised baselines on standard VI-ReID datasets.

**Questions:**

Please kindly refer to the weakness.

**Ethical Concerns:**

["NO or VERY MINOR ethics concerns only"]

**Final Justification:**

The authors' response has clarified my questions regarding the use of attribute information in the methodology and the model's computational overhead. Employing LVLM to enhance the performance of the VI-ReID task is an innovative approach and merits acceptance.

**Limitations:**

yes

**Quality:**

3

**Strengths And Weaknesses:**

Strength:
1. This work explores the highly novel direction by utilizing attribute arrays extracted by LVLM to improve the quality of cluster-based pseudo-labels in UVI-ReID, and designs an explicit-implicit attribute fusion module and an attribute-aware contrastive learning module based on attribute-aware strategies.
2. The proposed LVLM-AAM method significantly enhances the performance of UVI-ReID models. Extensive experiments are conducted to validate the effectiveness.

Weakness:
1. The explanation of the attribute-aware reliable labeling strategy in Section 3.2 requires more detail for full clarity. Specifically, it is unclear why attribute-aware matching selects only the first three attributes from the array, while attribute-aware refinement marks samples as outliers based on deviations exceeding η for all attributes. Does this imply different sensitivities or roles for attributes in these two sub-tasks?
2. The quality and reliability of the attributes extracted by the LVLM are crucial to the method's success, yet insufficiently analyzed. How effective is the LVLM at extracting meaningful attributes specifically from infrared images, given that LVLMs are typically pre-trained predominantly on visible spectrum data? Is there consistency in the attributes extracted for the same person across the visible and infrared modalities?
3. LVLM inference is computationally intensive. Integrating LVLM attribute extraction for every image and cluster within the ReID pipeline likely incurs significant time and resource overhead.
4. The relationship and training sequence between the LVLM-derived explicit attributes and the CLIP-learned implicit attributes need clarification: are the implicit attributes (learned via CLIP) acquired before applying the LVLM-AAM framework, as in previous similar works, or are they trained simultaneously?
5. The formulation of Equation (1) in Section 3.3 appears potentially incorrect due to the lack of normalization by the number of samples in the cluster.

---

> ### Author Rebuttal · Authors · 2025-07-29
>
> We thank the reviewer for the constructive and detailed feedback.
>
> ---
>
> > *Weakness1 and Question1: The explanation of the attribute-aware reliable labeling strategy in Section 3.2 requires more detail for full clarity. Specifically, it is unclear why attribute-aware matching selects only the first three attributes from the array, while attribute-aware refinement marks samples as outliers based on deviations exceeding η for all attributes. Does this imply different sensitivities or roles for attributes in these two sub-tasks?*
>
> **Response:**
> Thank you very much for your insightful comments. We use all attributes in the attribute-aware refinement module for outlier detection to align with a realistic perceptual principle: evaluating an object from more dimensions (attributes) is generally more accurate and comprehensive than doing so from fewer dimensions. Therefore, the attribute-aware refinement leverages all available attributes.
>
> For attribute-aware matching, our initial intention is also to use all available attributes to guide the grouping of clusters. However, we find this impractical in real-world scenarios. This is because clothing descriptions are highly diverse, and LVLM exhibits a certain degree of instability, which leads to varied interpretations of attributes such as “Upper” and “Lower.” For example, the same “Upper” item may be recognized by LVLM as either a “Shirt” or a “Sweatshirt,” resulting in broad value ranges for both “Upper” and “Lower.” If all five attributes are used for cluster grouping in attribute-aware matching, the resulting partitioning becomes overly fine-grained, often leading to groups that contain only visible clusters or only infrared clusters. This negatively impacts the inter-modality matching process. In contrast, the first three attributes in the attribute array—“Gender: [male/female],” “Glasses: [wearing/without],” and “Backpack: [carrying/without],”—have only two possible values each. By grouping clusters based on these three attributes, we are able to effectively leverage the knowledge provided by the LVLM while ensuring that each group contains both visible and infrared clusters. Therefore, as stated in lines 163-165 of the manuscript, “we choose the first three attributes because each has a countable value space, which ensures that each group contains both visible and infrared clusters.”
>
> Moreover, since the attribute-aware refinement only considers images with deviations exceeding $\eta$ (where $\eta = 2$) as outliers, it demonstrates a certain degree of tolerance and robustness to the variability or inaccuracy in the values of attributes such as “Upper” and “Lower.” As a result, it is feasible to utilize all five attributes in the attribute-aware refinement process.
>
> Therefore, as you correctly point out, the attributes indeed play different roles in the two modules, and attribute-aware matching is generally more sensitive to attribute variability than attribute-aware refinement. We will follow your suggestion to revise Section 3.2 to improve its clarity.
>
> ---
>
> > *Weakness2 and Question2: The quality and reliability of the attributes extracted by the LVLM are crucial to the method's success, yet insufficiently analyzed. How effective is the LVLM at extracting meaningful attributes specifically from infrared images, given that LVLMs are typically pre-trained predominantly on visible spectrum data? Is there consistency in the attributes extracted for the same person across the visible and infrared modalities?*
>
> **Response:**
> Thank you for your question. We take three main measures to ensure that the attributes extracted by the LVLM are effective for the ReID task. First, as shown in the supplementary material (Section S.I), we explicitly include the instruction “Exclude all color descriptions” when prompting the LVLM, to guide it to ignore color differences between visible and infrared images. Second, we evaluate several versions of the LVLMs, including Qwen2.5-VL-3B-Instruct, Qwen2.5-VL-7B-Instruct, Qwen2.5-VL-32B-Instruct, and Qwen2.5-VL-72B-Instruct. Balancing performance and computational cost, we choose Qwen2.5-VL-32B-Instruct, which provides a good trade-off, to extract image attributes and ensure their reliability. Third, in the attribute-aware matching module, we incorporate only the attributes “Gender,” “Glasses,” and “Backpack” during the inter-modality matching stage. These three attributes are typically invariant to the style shift caused by the infrared modality. Moreover, we define a compact value space for them—“Gender: [male/female],” “Glasses: [wearing/without],” and “Backpack: [carrying/without]”—to encourage consistency of these attributes across modalities.
>
> Based on the above three efforts, we observe that the inconsistency rate of the first three attributes for the same person across the visible and infrared modalities is less than 7%.
>
> ---
>
> > *Weakness3 and Question3: LVLM inference is computationally intensive. Integrating LVLM attribute extraction for every image and cluster within the ReID pipeline likely incurs significant time and resource overhead.*
>
> **Response:**
> Thank you very much for your comment. We acknowledge that LVLM inference is computationally intensive. For our task, the Qwen2.5-VL-32B-Instruct model is sufficient to improve the performance of our ReID framework without incurring prohibitive costs. Specifically, Qwen2.5-VL-32B-Instruct runs efficiently on our ReID training hardware (i.e., four NVIDIA GeForce RTX 4090 GPUs), and the inference time is generally acceptable. For example, extracting image-level attributes for all training images in the SYSU-MM01 dataset typically takes around 15 hours. Moreover, obtaining cluster-level attribute arrays does not involve additional LVLM inference. Instead, it only requires computing the mode of all image-level attribute arrays within a cluster, which is a very fast operation. This process usually takes only a few seconds for the entire dataset.
>
> In addition, the LVLM inference is only required once during the training phase to extract identity attributes and is not needed during the testing phase. Therefore, the computational cost associated with LVLM inference does not affect the inference speed of the trained ReID model during testing.
>
> ---
>
> > *Weakness4 and Question4: The relationship and training sequence between the LVLM-derived explicit attributes and the CLIP-learned implicit attributes need clarification: are the implicit attributes (learned via CLIP) acquired before applying the LVLM-AAM framework, as in previous similar works, or are they trained simultaneously?*
>
> **Response:**
> Thank you for your question. As described in lines 195-197 of the manuscript, we first combine the LVLM-derived explicit attributes with the implicit attributes by assigning to each cluster a text prompt containing learnable text embeddings “$\left[ {{X_1}} \right]{\rm{ }}\left[ {{X_2}} \right] \ldots \ldots \left[ {{X_M}} \right]$” in the format:
> “An image of a $\left[ {{X_1}} \right]{\rm{ }}\left[ {{X_2}} \right] \ldots \ldots \left[ {{X_M}} \right]$ $[male/female]$ $[wearing/without]$ glasses and $[carrying/without]$ a backpack.” Subsequently, we introduce the CLIP contrastive loss (as defined in Equation (6)) to optimize the learnable text embeddings.
>
> Therefore, the explicit and implicit attributes are jointly fed into the model. In addition, only the implicit attributes (i.e., the learnable text embeddings) are optimized, while the explicit attributes remain fixed throughout training.
>
> ---
>
> > *Weakness5 and Question5: The formulation of Equation (1) in Section 3.3 appears potentially incorrect due to the lack of normalization by the number of samples in the cluster.*
>
> **Response:**
> We apologize for the confusion and would like to provide further clarification regarding Equation (1). Since the number of samples within each cluster may vary, we use $N_p^v$ to denote the number of images (samples) in the $p$-th cluster. The subscript of $N_p^v$ corresponds directly to the cluster index $p$. Therefore, in Equation (1), each cluster is associated with its own intra-cluster sample count.

---

> > ### Comment · Reviewer_nERK · 2025-08-05
> >
> > Thank you for your detailed response, which has clarified my questions regarding the use of attribute information in the methodology and the model's computational overhead. Employing LVLM to enhance the performance of the VI-ReID task is an innovative approach and merits acceptance.

---

> > > ### Author Response · Authors · 2025-08-05
> > >
> > > Thank you for your positive feedback and recognition. We greatly appreciate your support, which is very encouraging for our work.

---

### Note · Authors · 2025-08-15

We sincerely thank the reviewers for their constructive feedback and recognition, which are invaluable for both our current and future work. We also appreciate the Area Chair’s time and effort.

We summarize the contributions and novelty of this work as follows:

This paper introduces an LVLM-AAM method to improve visible-infrared person re-identification. The key difference from existing approaches lies in LVLM-AAM’s use of supervisory information (i.e., attribute arrays) from a large vision-language model (LVLM) to refine pseudo-labels and text semantics, thereby enhancing modality-invariant feature learning.

Within LVLM-AAM, we propose attribute-aware refinement (AR) and attribute-aware matching (AM) to refine intra-modality and inter-modality pseudo-labels, respectively. In addition, we develop an explicit-implicit attribute fusion (EAF) module, which fuses implicit attributes (text embeddings) and explicit attributes (attribute arrays) to produce more fine-grained, identity-related text features. Finally, we introduce an attribute-aware contrastive learning (AAC) module, which computes dynamic text features based on static text features from both modalities, and optimizes using both static and dynamic features to further enhance modality-invariant learning.

Once again, we thank the reviewers and the Area Chair, and we will revise the paper accordingly based on the reviewers’ suggestions.

---

### Decision · Program_Chairs · 2025-09-17

**Decision:**

Accept (poster)

**Comment:**

This paper proposes an attribute-aware modeling method LVLM-AAM for unsupervised visible-infrared person re-identification, which uses a large vision-language model to extract attributes, and enhances modality-invariant learning. Experimental results on two datasets demonstrate the superiority of the proposed method.

This paper constructs an intersting method and achieve good performances, all reviewers give positive scores and their concerns are well addressed during the discussion phase. After complementing the statements and experiments provided in the discussion phase, this paper is expected to reach the acceptance level of NeurIPS 2025.